# D2K: Turning Historical Data into Retrievable Knowledge for Recommender Systems

## ABSTRACT

A vast amount of user behavior data is constantly accumulating on today's large recommendation platforms, recording users' various interests and tastes. Preserving knowledge from the old data while new data continually arrives is a vital problem for recommender systems. Existing approaches generally seek to save the knowledge implicitly in the model parameters. However, such a parameter-centric approach lacks scalability and flexibility—the capacity is hard to scale, and the knowledge is inflexible to utilize. Hence, in this work, we propose a framework that turns massive user behavior data to retrievable knowledge (D2K). It is a *data-centric* approach that is model-agnostic and easy to scale up. Different from only storing unary knowledge such as the user-side or item-side information, D2K proposes to store *ternary knowledge* for recommendation, which is determined by the complete recommendation factors—user, item, and context. The knowledge retrieved by target samples can be directly used to enhance the performance of any recommendation algorithms. Specifically, we introduce a Transformer-based knowledge encoder to transform the old data into knowledge with the user-item-context cross features. A personalized knowledge adaptation unit is devised to effectively exploit the information from the knowledge base by adapting the retrieved knowledge to the target samples. Extensive experiments on two public datasets show that D2K significantly outperforms existing baselines and is compatible with a major collection of recommendation algorithms.

## 1 INTRODUCTION

Real-world recommender systems accumulate a substantial volume of user logs every day [22]. It makes training a recommender with all the data intractable since the computational resources are limited. However, using only the recent logs by truncating the data to a fixed time window is suboptimal because valuable information may be abandoned together with the old data [19, 20]. Hence, the major problem is *preserving useful knowledge from the old data effectively*.

Most existing studies in recommendation scenarios [15, 28, 30, 32] consider preserving the knowledge in model parameters implicitly using continual learning techniques. We refer to it as *parameter-centric knowledge*. As shown in Figure 1(a), the key point of the parameter-centric approach is inheriting knowledge from the old model trained on old data. Knowledge distillation [28, 30] is usually utilized to transfer the knowledge to the new model with the old one acting as the teacher. Memory-augmented networks are introduced to enhance the memorization capacity of the model by augmenting the networks with an external memory component [8, 9, 12]. Additional information is stored in the external memory, such as user interest representations [4, 26], or knowledge graph-enhanced item representations [11]. The number of parameters for each user/item is expanded so that more information can be memorized. Meta-learning is also explored in continual learning for recommendation [32] by learning to optimize future performance.

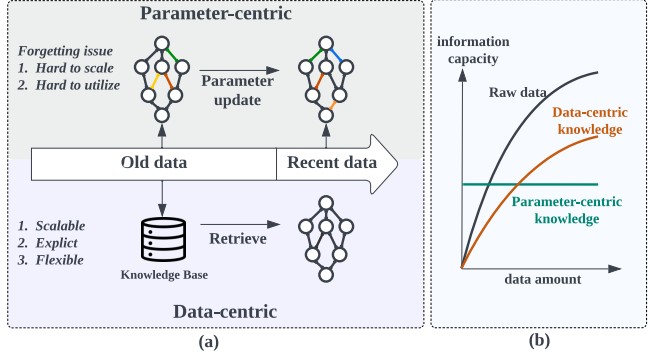

**Figure 1: Parameter-centric knowledge vs. data-centric knowledge in preserving information. (a) Workflow comparison. (b) Conceptual illustration of information capacity.**

Nevertheless, the long-standing *catastrophic forgetting* problem still limits the performance and usability of the parameter-centric knowledge. Catastrophic forgetting means the knowledge in the old data is preserved in the model parameters, and updating the model with new data interferes with previously learned knowledge [16]. We argue that there are two major reasons for catastrophic forgetting. The first is that the memorization capacity could not scale up with the data size. The number of model parameters is fixed while the user logs keep growing. When adding new information into the fixed-size parameters, we have to erase some of the stored information and insert the new information, inevitably resulting in information loss. As shown in Figure 1(b), the raw data contains the full information and has the largest information capacity. The parameter-centric methods transform the raw data into a fixed number of parameters and thereby cannot scale up as the data size increases, resulting in the loss of information. The second reason for the forgetting issue is that the stored knowledge in parameters is difficult to utilize appropriately. It is always hard to manage the trade-off between the old and newly learned knowledge [29].

It could be inevitable for a model to suffer from catastrophic forgetting, so we try a data-centric approach that turn the historical data into retrievable knowledge as shown in Figure 1(a). *The retrievable knowledge could help the model to recollect the old but useful patterns.* The strengths of the data-centric knowledge base are in three folds: (i) **Scalability**. The number of entries in the knowledge base can grow with the data amount. For the knowledge base, adding new information is a simple process of inserting new entries instead of the complex process of updating model parameters, which is more scalable as presented in Figure 1(b). (ii) **Explicitness**. We could store explicit knowledge, which will act as additional features to enhance prediction performance. It is easier to utilize compared to the implicit knowledge in model parameters. (iii) **Flexibility**. A knowledge base is model-agnostic. Thus, it is compatible with different backbone models.

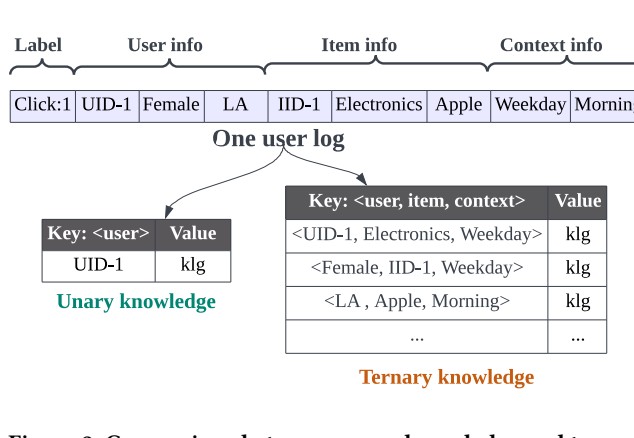

**Figure 2: Comparison between unary knowledge and ternary knowledge. "klg" stands for knowledge.**

Despite the desirable properties of the data-centric approach, it is still challenging to determine what type of knowledge should be extracted from the data and to be stored. As shown in Figure 2, the log means a female user (UID-1) who lives in LA has clicked an Apple electronic device (IID-1) on a weekday morning. To preserve information in that log, unary knowledge format is always utilized—the knowledge is indexed by a unary term such as the user ID [17, 22]. Thus the log data could only provide one signal that the user has an interest in the item or the user interest is updated according to the item. We believe that the user feedback, positive or negative, is affected by three types of information together, i.e., user information, item information, and context information. These three aspects interact with each other and affect user feedback [1, 10, 13, 21]. Therefore, the knowledge should be in a ternary form that contains all user-item-context information. As shown in Figure 2, we want the log data to tell us how the ternary cross feature <UID-1, Electronics, Weekday>[1] affects the label. By listing all the ternary features from the log, we are capable of extracting more fine-grained and abundant knowledge. As such, the stored knowledge directly records the underlying causes and motives of a particular recommendation outcome (click or not).

In this paper, we propose a framework that turns massive user behavior data to retrievable knowledge (D2K). D2K aims to transform the old data that is not included in the training set into a key-value style knowledge base. Ternary knowledge is extracted from the log data and stored in the knowledge base. The knowledge will be retrieved by target samples to enhance the performance of any recommendation algorithms. Specifically, in D2K, there are two major components. The first is a Transformer-based *knowledge encoder* that encodes any ternary cross features into a knowledge vector. All the unique ternary features in the old data will be used as keys in the knowledge base, and the values will be given by the knowledge encoder. The second is a *personalized knowledge adaptation unit* which is leveraged when utilizing the information from the knowledge base. The global knowledge is adapted to the current target sample through a neural network whose parameters come from the input target sample. Our main contributions are summarized as three folds:

---

[1]For simplicity, we use one feature for each aspect.

- For the problem of preserving information from massive user logs, we are the first to provide a data-centric approach by building a knowledge base. It could be a better choice for recommendation scenarios than the parameter-centric approaches.
- We are the first to propose a ternary knowledge base for recommendation to transform the old data into retrievable knowledge. Ternary knowledge provides more fine-grained and abundant information compared to traditional unary knowledge.
- We design a Transformer-based knowledge encoder to effectively extract knowledge for any given ternary features and a personalized knowledge adaptation unit to adapt the knowledge to specific target samples without introducing much more parameters.

## 2 METHODOLOGY

In this section, we first introduce the preliminaries and give a big picture of the framework, followed by detailed explanations of D2K's knowledge generation and knowledge utilization procedures.

### 2.1 Preliminaries

In this section, we formulate the problem and describe the notations. The user behavior log is the core data type of the recommendation scenario. Each log data point is denoted as $(x, y)$ where $x = \{u, v, c\}$. The data sample represents one user behavior that user $u$ has interacted with item $v$ under context $c$ with the feedback label $y$.

There are multiple feature fields in each data point. The user features are denoted as $u = \{f_i^u\}_{i=1}^{F_u}$, item features are $v = \{f_j^v\}_{j=1}^{F_v}$ and context features are $c = \{f_k^c\}_{k=1}^{F_c}$. $F_u, F_v, F_c$ denote the number of the features for the user, the item, and the context, respectively. The features could be categorical or numerical, and they can be either single-value (such as gender, category, etc) or multi-value (such as user historical sequence of clicked items).

In our proposed D2K framework, we argue that retrieving knowledge of the target sample from the old data could benefit the prediction. Thus the estimation function is formulated as

$$\hat{y} = f_\Theta(u, v, c, \mathcal{R}_\Phi(u, v, c)). \tag{1}$$

$\mathcal{R}_\Phi$ represents the knowledge retrieval and utilization procedure from the old data, which is parameterized by $\Phi$ and $f_\Theta$ is the learned scoring function.

### 2.2 D2K Framework Overlook

The overall framework is shown in Figure 3. The entire dataset could be divided into two parts according to timestamp: old data and recent data. The recent data corresponds to the normal train and test data. We aim to extract and preserve useful knowledge from the old data to help the prediction on the recent data samples. Figure 3 provides an overlook of the D2K framework, which consists of two parts: knowledge generation and knowledge utilization. In the knowledge generation process, the old data is transformed into a ternary knowledge base via a knowledge encoder. In the knowledge utilization process, for each target sample in the recent data, we generate user-item-context features as the query to lookup the corresponding knowledge from the knowledge base. The retrieved knowledge is then adapted and injected into an arbitrary recommendation model (RS model).

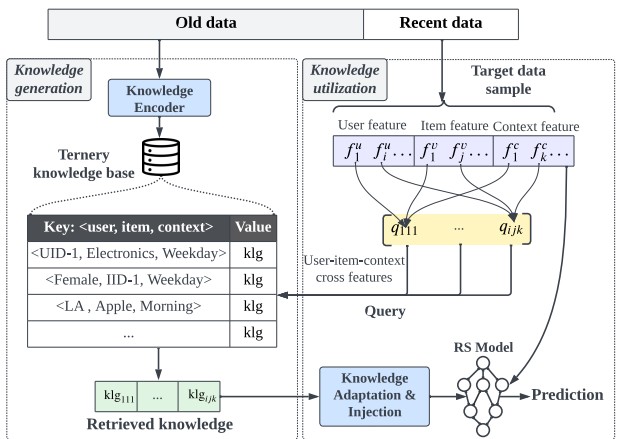

**Figure 3: The framework of D2K.**

## 2.3 Knowledge Generation

In this section, we first introduce the ternary knowledge with its design motivations. Then we present the knowledge encoder and the procedure to transform the old data into a knowledge base.

*2.3.1 Ternary Knowledge.* The major design of D2K is utilizing ternary tuples as the keys of the knowledge base, as shown in Figure 3. One user behavior is affected by three aspects: user aspect, item aspect, and context aspect. These three aspects interact with each other and indicate whether the clicking event will happen. Unlike the unary knowledge widely used in memory-augmented networks, ternary knowledge is "direct" knowledge that could be used solely to produce a prediction as described in Definition 1.

**DEFINITION 1 (DIRECT KNOWLEDGE).** *A knowledge vector $z_x$ retrieved by the sample $x$ is defined as direct knowledge if it carries enough information to produce a prediction on sample $x$ as $func(z_x) \xrightarrow{produce} \hat{y}$.*

Take the ternary keys in Figure 3 as an example, the knowledge of "<UID-1, Electronics, Weekdays>" has all three aspects of a clicking event (user aspect, item aspect, and context aspect). Thus it could be directly used to produce a prediction solely[2]. On the contrary, unary knowledge is not able to produce a prediction directly. For example, if the knowledge is only the interest representation vector of the target user (use user ID to retrieve), the knowledge alone could not produce a prediction. It has to be combined with the target item and context information to produce a clicking estimation. Therefore, we extract and preserve the ternary knowledge in D2K as it carries the direct information of clicking events.

*2.3.2 Knowledge Encoder.* The crucial problem of D2K is how to encode the knowledge of each ternary key. We propose a Transformer-based [24] knowledge encoder as shown in Figure 4. For ease of expression, we omit the $u, i, c$ feature source and denote the input

---

[2]The knowledge could be used together with the input $x$ for better performance, we only want to illustrate the strength of the direct knowledge.

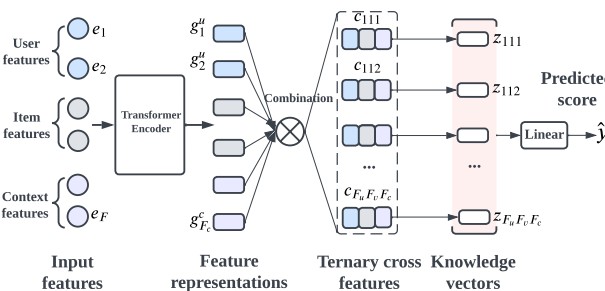

**Figure 4: The structure of the knowledge encoder of D2K.**

sample $x = \{f_i\}_{i=1}^{F}$ with $F$ features. We define that

$$Q = K = V = \begin{pmatrix} e_1 \\ \vdots \\ e_F \end{pmatrix}, \tag{2}$$

where $e_i$ is the embedding of the $i$-th feature in $x$. Note that feature $f_i$ could be either a single-value or multiple-value feature (such as user-clicked items). If feature $f_i$ has multiple values, the embedding $e_i$ is the average pooling of the elements of $f_i$ as $e_i = \frac{1}{n_i}\sum_{p=1}^{n_i} b_p$, where $b_p$ is the embedding of each feature element and $n_i$ is the number of feature values of feature field $i$. The output of the Transformer-based encoder is

$$E = \text{TRM}(Q, K, V), \tag{3}$$

where TRM represents Transformer. And the resulting matrix $E \in \mathbb{R}^{F \times d}$ could be regarded as stacked representations from three sources of user, item, or context as $\{g_1^u, ..., g_{F_u}^u, g_1^v, ..., g_{F_v}^v, g_1^c, ..., g_{F_c}^c\}$. We use $g$ to denote the feature representations. The Transformer is chosen as the major component because of its superior modeling capacity and flexibility w.r.t. the input length. The Transformer could take input of various lengths, which is especially useful because the number of input features differs in the encoder training ($F$ features) and the knowledge extracting process of D2K (3 features).

To extract the knowledge of each ternary cross feature, we generate all the combinations of the user-item-context cross features. The cross features are described as

$$c_{ijk} = \text{concat}(g_i^u, g_j^v, g_k^c), \ \forall i \in [1, F_u], j \in [1, F_v], k \in [1, F_c]. \tag{4}$$

Each ternary cross feature $c_{ijk}$ is further input to a knowledge network and gets the corresponding knowledge vector as

$$z_{ijk} = \text{MLP}(c_{ijk}), \ \forall i \in [1, F_u], j \in [1, F_v], k \in [1, F_c], \tag{5}$$

where $z_{ijk} \in \mathbb{R}^{d_k}$ represents the knowledge extracted for the ternary key $\langle f_i^u, f_j^v, f_k^c \rangle$ and $d_k$ is the length of the knowledge vector. After obtaining the knowledge vector of each ternary cross feature, we concatenate them and feed them into a linear layer as

$$\hat{y} = \sigma(w_l^T \cdot (\text{concat}(z_{111}, ..., z_{F_u F_v F_c})) + b_l), \tag{6}$$

where $w_l \in \mathbb{R}^{F_u F_v F_c d_k \times 1}$ is the weight vector, $b_l$ is the bias and $\sigma(x) = \frac{1}{1+e^{-x}}$ is the sigmoid function. This way, the knowledge vector $z_{ijk}$ could be regarded as linear distinguishable. In other words, the Transformer and MLP layers map the knowledge into a linearly separable space.

The encoder is trained using binary cross-entropy loss as

$$\mathcal{L}_{enc} = -\sum_{(x,y)\in\mathcal{D}_{old}} y \cdot \log(\hat{y}) + (1-y) \cdot \log(1-\hat{y}), \quad (7)$$

where $\mathcal{D}_{old}$ represents the old data shown in Figure 3.

*2.3.3 Process of the Knowledge Base Generation.* After the encoder is trained, the Transformer and the knowledge network are then fixed and used to generate knowledge of any given ternary cross feature. All the ternary cross features that appear in $\mathcal{D}_{old}$ will be fed into the encoder to derive the corresponding knowledge representations. The detailed process for generating the knowledge base $\mathcal{K}$ is shown in Algorithm 1.

For the feature with multiple values, it will be split and treated as multiple single values. Say $f_i^u = \{b_p^u\}_{p=1}^{n_i}$, then there are $n_i$ tuples for the ternary features of $\langle f_i^u, f_j^v, f_k^c \rangle$, which are $\{\langle b_p^u, f_j^v, f_k^c \rangle\}_{p=1}^{n_i}$.

*2.3.4 Knowledge Update.* As the historical data grows, the knowledge should be updated in the knowledge base. We will train a new encoder to handle the new data that exceeds the current encoder's capacity. For example, each encoder will be responsible for 7-day logs, the subsequent historical logs will be used to train a new encoder and these logs will be turned into knowledge via the process shown in Algorithm 1.

If the current knowledge base does not contain the ternary key, the new entry will be directly inserted into the base. Otherwise, we use different approaches: (1) Recent Priority (RP), the old knowledge vector will be directly replaced by the new one. (2) Average Pooling (AP), the new knowledge vector and the old vectors will be averaged.

## 2.4 Knowledge Utilization

*2.4.1 Query Generation.* As illustrated in Figure 3, given a target data sample, we convert it into a query set to retrieve the corresponding knowledge. As the keys are in the ternary format, the query terms should also be ternary. For a given target sample $x$, we derive its query set $Q_x$ as,

$$\begin{aligned} Q_x = \{q_t\}_{t=1}^{N_q} = \{\langle f_i^u, f_j^v, f_k^c \rangle\}, \\ \forall i \in [1, F_u], j \in [1, F_v], k \in [1, F_c], \end{aligned} \quad (8)$$

where $N_q = F_u \times F_v \times F_c$ is the total number of query terms. All the query terms in $Q_x$ represent the complete information we want from the knowledge base for sample $x$.

*2.4.2 Knowledge Lookup.* The query set $Q_x$ will be used to retrieve the knowledge by looking up the corresponding keys in the knowledge base $\mathcal{K}$. The retrieved knowledge of the target sample $x$ is $[z_{111}, ..., z_{ijk}, ..., z_{F_u F_v F_c}]$. If a query $q_t$ in $Q_x$ involves multi-value features, it will be split into several single-value features, and the retrieved knowledge vectors will be averaged. For example, if in $q_t = \langle f_i^u, f_j^v, f_k^c \rangle$, $f_i^u$ is a multi-value feature that $f_i^u = \{b_p^u\}_{p=1}^{n_i}$, then the knowledge vector of $q_t$ is

$$z_{ijk} = \frac{1}{n_i} \sum_{p=1}^{n_i} z_{pjk}, \quad (9)$$

where $z_{pjk}$ is the retrieved knowledge of $\{\langle b_p^u, f_j^v, f_k^c \rangle\}$. It should be noticed that the retrieved knowledge vectors in $\mathcal{K}$ are all constant

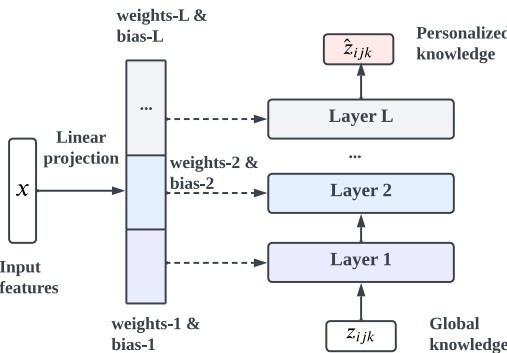

**Figure 5: The structure of the personalized knowledge adaptation unit.**

values. Their values will not be updated in the training process afterwards.

*2.4.3 Knowledge Adaptation.* The knowledge we acquired from the previous section is actually the "global" knowledge because the encoder is trained on the entire old dataset $\mathcal{D}_{old}$. Thus the knowledge of each ternary key reflects a global and average knowledge of a certain user-item-context feature triplet. For example, the knowledge of the ternary tuple "LA (user location) & Apple (item category) & Morning (context)" reflects the average influence on the clicking probability of these three features interacting with each other. It is global knowledge because it is learned from all the samples containing these ternary features. We want to make the knowledge "personalized" w.r.t the target sample $x$.

As shown in Figure 5, we propose the personalized knowledge adaptation unit. The main idea of the personalized knowledge adaptation unit is to use the target sample $x$ as *MLP parameters* [2, 6]. The corresponding model parameters in the MLP are generated by a specific target sample $x$. Thus for different $x$, the MLP will be different. The knowledge vector is fed forward through the $x$-personalized MLP and gets the personalized knowledge representation w.r.t $x$. This is a parameter-efficient and personalized way of adapting global knowledge.

The input vector is $x = [e_1, ..., e_F]$. A linear projection layer is first conducted to map the input to a suitable shape as

$$w_x = w_{pro} \cdot x, \quad (10)$$

where $w_x \in \mathbb{R}^{(Ld_k(d_k+1))}$, $w_{pro} \in \mathbb{R}^{Ld_k(d_k+1)\times Fd}$ and $x \in \mathbb{R}^{Fd}$. $L$ represents the number of layers in the adaptation MLP. As shown in Figure 5, the weights and biases of each MLP layer are sliced and reshaped from the projected input $w_x$. We have

$$w_x = \text{concat}(\{w_x^l, b_x^l\}_{l=1}^{L}), \quad (11)$$

where $w_x^l$ will be reshaped as $w_x^l \in \mathbb{R}^{d_k \times d_k}$ and $b_x^l \in \mathbb{R}^{d_k}$. The personalized knowledge $\hat{z}_{ijk}$ is calculated as

$$\hat{z}_{ijk} = \delta_L(w_x^L(\ldots \delta_1(w_x^1 \cdot z_{ijk} + b_x^1) \ldots) + b_x^L), \quad (12)$$

where $\hat{z}_{ijk} \in \mathbb{R}^{d_k}$ and $\delta_l$ is the activation function of layer $l$. We use Tanh as the activation function that $\delta_l(x) = \frac{e^x - e^{-x}}{e^x + e^{-x}}$.

*2.4.4 Knowledge Injection.* For injecting the knowledge to an arbitrary deep recommendation model $\mathcal{M}$, we propose two different ways: concatenation and separate tower. The first way is to concatenate the knowledge vector with the original input as

$$\hat{x} = \text{concat}(x, \{\hat{z}_{ijk}\}). \tag{13}$$

The knowledge-augmented input $\hat{x} \in \mathbb{R}^{Fd+N_q d_k}$ will be fed into the deep feed-forward network component of $\mathcal{M}$. This will change the shape of the parameter matrix of $\mathcal{M}$.

The second way is incorporating the knowledge by adding a separate tower to produce the final estimation as

$$\hat{y} = \sigma(\mathcal{M}(x) + \mathcal{P}(\text{concat}(\{\hat{z}_{ijk}\}))), \tag{14}$$

where $\mathcal{P}$ is the knowledge predictor which could be implemented as either a linear function or a deep neural network. Adding a separate tower will not affect the original structure of $\mathcal{M}$ thus it is flexible and easier for implementation. Lastly, the knowledge-enhanced recommendation model is trained using binary cross entropy loss.

# 3 EXPERIMENTS

In this section, we present the experimental results and corresponding analysis. The implementation code is made public[3]. Four research questions (RQ) lead the following discussions.

- **RQ1**: Does D2K achieves best performance?
- **RQ2**: Is every component of D2K effective and essential?
- **RQ3**: How to reduce the size of the knowledge base? What is the effect of using a smaller knowledge base?
- **RQ4**: What is the performance of knowledge update methods?

## 3.1 Experimental Settings

*3.1.1 Datasets.* To verify the effectiveness of the D2K framework, we use two large-scale public datasets from real-world scenarios: AD and Eleme. Statistics can be found in Appendix A.2.

- **AD**[4] is a displayed advertisement dataset provided by Taobao, a large online shopping platform in China. It contains the exposure and clicking logs of over one million users from 20170506 to 20170512. Apart from the ordinary user/item/context features, it also provides the sequential behaviors in the previous 22 days of each user.
- **Eleme**[5] is constructed by click logs from Eleme online recommendation system. Eleme mainly provides food takeouts to users. It provides abundant features which are valuable to D2K.

**Dataset Partition.** All the samples are sorted in chronological order first. And we use a fixed time window (e.g., a day) to split the data into several blocks as $\mathcal{D}_1, ..., \mathcal{D}_T$. The old data $\mathcal{D}_{old}$ is formed as $\mathcal{D}_{old} = \{\mathcal{D}_1, ..., \mathcal{D}_{p_1}\}$, training set is formed as $\mathcal{D}_{tr} = \{\mathcal{D}_{p_1+1}, ..., \mathcal{D}_{p_2}\}$ and the remaining blocks form the test set $\mathcal{D}_{te} = \{\mathcal{D}_{p_2+1}, ..., \mathcal{D}_T\}$, where $p_1$ and $p_2$ are partition points.

*3.1.2 Compared Methods.* As the proposed D2K is a framework compatible with arbitrary recommendation models, we choose **DeepFM** [10], **DIN** [33] and **DCNv2** [27] as the backbones of our experiments because they are widely used models. The compared

---
[3]https://bit.ly/40bzoPV
[4]https://tianchi.aliyun.com/dataset/56
[5]https://tianchi.aliyun.com/dataset/131047

methods could be used to preserve knowledge from the old data and are all compatible with any recommendation model.

- **Fixed Window** is the most basic method. It uses the data in a fixed-size of time window as the training data directly. For this method, we test two different variants. The first one only uses recent data $\mathcal{D}_{tr}$ as the training set (Fixed Window (R)), and the second one uses all data as the training set as $\{\mathcal{D}_{old}, \mathcal{D}_{tr}\}$ (Fixed Window (A)). The second variant is not practical in the real world. We test it because it contains the maximum information we could possibly get for our offline datasets.
- **Incremental** is the simple incremental update baseline. It iteratively uses $\mathcal{D}_1, ..., \mathcal{D}_{p_2}$ to train the recommendation model. The next data block will not be used until the model converges on the previous data block. It's widely used in real world [28].
- **IncCTR** [28] is an advanced method for incremental learning. The major improvement of IncCTR is using the old model trained on old data as a teacher to teach the model trained on the new data. We use the knowledge distillation variant of IncCTR (KD-batch) because its performance is better.
- **Pretrain embedding** uses $\mathcal{D}_{old}$ to pretrain a model. The new model will inherit the embedding table of the pretrained model as the initialization of its embeddings. The new model is then trained on $\mathcal{D}_{tr}$. We use the same model structure as the three backbone models respectively in the pre-training.
- **MemoryNet** represents the unary knowledge in external memory. The memory's key is the user ID, and the value is a learned user interest representation using her sequential behaviors. The network structure to capture the user interest is HPMN [22].
- **Random** is a simple coreset selection method that uses random sampling to select some of the data samples from $\mathcal{D}_{old}$ so as to compress it. Thus the training data is regarded as $\mathcal{D}_{tr}$+Zip($\mathcal{D}_{old}$). The selected coreset size is 10% of $\mathcal{D}_{old}$.
- **SVP-CF** [23] is a coreset selection method specifically proposed for recommender systems. It uses a proxy model to select the samples that the proxy predicts with the largest deviation from the ground truth. These most difficult samples are regarded as the coreset. The selected coreset size is 10% of $\mathcal{D}_{old}$.

We then introduce four different D2K variants in the following:

- **D2K-base** is the basic version of D2K framework without the personalized knowledge adaptation unit.
- **D2K-adp-sep** has the personalized knowledge adaptation unit, but it uses a separate embedding table for the input $x$ when used as the MLP parameters. It means the input sample $x$ has two different embeddings, one for ordinary use and another for knowledge adaptation. We use separate embeddings to avoid these two roles of $x$ affecting each other. The separate embedding has the same size as the original one.
- **D2K-adp-small** differs with D2K-adp-sep only on the embedding size for the adaptation unit. It uses a 1/4 embedding size for input $x$ in the adaptation unit to reduce parameters.
- **D2K-adp-share** uses share embedding of input $x$ for the adaptation unit. It is exactly the model that we propose in Section 2.

## 3.2 Overall Performance (RQ1)

In this section, we compare the performance of D2K with other parameter-centric approaches and the coreset methods. The overall

performance of D2K is shown in Table 1 and Table 2. The reported results are the average performance of five trials with different random seeds. We have the following observations:

- For both datasets and backbone models, D2K-adp-share (blue shaded) performs better than all the baseline methods in terms of AUC. The AUC improvement rates against the best baselines are 1.35%, 1.70%,1.52% on the AD dataset for the three backbone models, respectively. And the improvement rates on Eleme are 2.90%, 2.33%, and 1.89%. In fact, an improvement rate at 1%-level in AUC is likely to yield a significant online performance improvement [19, 21]. As for LL, D2K-adp-share performs better on AD but not as well on the Eleme dataset. For recommender systems, the item's relative ranking order (AUC) is the key metric, and is more important than the point-wise accuracy of the ranking score (LL). Hence, the results are satisfying in general.
- Compared with the Fixed Window (A) on all data $\mathcal{D}_{old} + \mathcal{D}_{tr}$ (gray shaded), D2K-adp-share outperforms it. Theoretically, the Fixed Window (A) on all data contains the maximum information of our offline datasets, whereas D2K inevitably has some loss of information. The results show that directly using all data as training samples may not be the best practice. Though with the maximum information, Fixed Window (A) is essentially parameter-centric and preserves the knowledge in the model parameters through gradient descent. Transforming the old data into ternary knowledge maintains all three important factors for recommendation, leading to superior performance.
- The two incremental learning (continual learning) baselines (Incremental & IncCTR) perform worse than Fixed Window (R) on the AD dataset but better than it on Eleme. Eleme dataset has more data, making it difficult to memorize the important patterns by only utilizing the recent data. AD dataset is smaller thus each data block may not cover enough samples for the incremental learning methods to be trained sufficiently. These reasons could explain why incremental learning performs worse on AD dataset. D2K-adp-share, on the contrary, is less affected by these issues and steadily outperforms the baselines.
- Pretrain embedding and MemoryNet could be seen as using unary knowledge to preserve the information in the old data. Pretrain embedding uses unique feature ids as keys and the corresponding pretrained embedding vectors as values. MemoryNet uses user ID as keys and user interest representations as knowledge. As D2K-base performs better than the two baselines, we could verify that the direct ternary knowledge stored in D2K provides more useful information to the recommendation models than the unary knowledge.
- The coreset selection baselines Random & SVP-CF are simple yet effective methods to preserve knowledge because, in most cases, they perform better than Fixed Window (R) on recent data. The coreset methods even produce better results than the other more complex methods on the AD dataset. However, the coreset methods are based heavily on heuristics such as pre-defined data selection criteria. And they drop a large portion of original data, which could lead to severe information loss.

The time & space overhead of D2K is shown in Appendix A.3. To further demonstrate the usefulness of D2K, we also test the performance of only using the "direct knowledge" (Definition 1) to produce predictions without original input $x$ in Appendix A.4.

**Table 1: Performance on AD. Improvements over baselines are statistically significant with $p < 0.05$.**

| Method | DeepFM | | DIN | | DCNv2 | |
|---|---|---|---|---|---|---|
| | AUC | LL | AUC | LL | AUC | LL |
| Fixed Window (R) | 0.6202 | 0.1952 | 0.6203 | 0.1949 | 0.6211 | 0.1952 |
| Incremental | 0.6188 | 0.1954 | 0.6183 | 0.1960 | 0.6192 | 0.1957 |
| IncCTR (KD-batch) | 0.6191 | 0.1955 | 0.6194 | 0.1955 | 0.6107 | 0.1973 |
| Pretrain embedding | 0.6202 | 0.1949 | 0.6209 | 0.1950 | 0.6212 | 0.1948 |
| MemoryNet | 0.6192 | 0.1952 | 0.6195 | 0.1959 | 0.6215 | 0.1956 |
| Random | 0.6223 | 0.1946 | 0.6233 | 0.1946 | 0.6240 | 0.1946 |
| SVP-CF | 0.6234 | 0.1946 | 0.6234 | 0.1946 | 0.6246 | 0.1945 |
| D2K-base | 0.6328 | 0.1940 | 0.6333 | 0.1938 | 0.6339 | **0.1937** |
| D2K-adp-sep | 0.6325 | **0.1938** | 0.6338 | 0.1938 | 0.6338 | 0.1938 |
| D2K-adp-small | **0.6330** | 0.1939 | 0.6339 | **0.1937** | 0.6337 | 0.1938 |
| D2K-adp-share | 0.6318 | 0.1946 | **0.6340** | 0.1939 | **0.6341** | 0.1940 |
| *Fixed Window (A)* | *0.6237* | *0.1944* | *0.6228* | *0.1945* | *0.6235* | *0.1944* |

**Table 2: Performance on Eleme. Improvements over baselines are statistically significant with $p < 0.05$.**

| Method | DeepFM | | DIN | | DCNv2 | |
|---|---|---|---|---|---|---|
| | AUC | LL | AUC | LL | AUC | LL |
| Fixed Window (R) | 0.5769 | 0.0987 | 0.5950 | 0.0913 | 0.5800 | 0.1356 |
| Incremental | 0.5863 | 0.0932 | 0.5968 | 0.0949 | 0.5857 | 0.0951 |
| IncCTR(KD-batch) | 0.5806 | **0.0897** | 0.5957 | 0.0917 | 0.6033 | **0.0910** |
| Pretrain embedding | 0.5775 | 0.0976 | 0.5883 | 0.0915 | 0.5724 | 0.1142 |
| MemoryNet | 0.5745 | 0.0994 | 0.5910 | 0.0911 | 0.5845 | 0.1302 |
| Random | 0.5763 | 0.0965 | 0.5959 | 0.0900 | 0.5817 | 0.0997 |
| SVP-CF | 0.5771 | 0.0962 | 0.5893 | 0.0899 | 0.5885 | 0.1031 |
| D2K-base | 0.5832 | 0.1031 | 0.5917 | **0.0892** | 0.5863 | 0.3256 |
| D2K-adp-sep | **0.6196** | 0.0955 | 0.6093 | 0.0901 | 0.6266 | 0.1193 |
| D2K-adp-small | 0.6176 | 0.1365 | 0.6009 | 0.0981 | **0.6401** | 0.1343 |
| D2K-adp-share | 0.6033 | 0.1081 | **0.6107** | 0.0905 | 0.6147 | 0.1913 |
| *Fixed Window (A)* | *0.5918* | *0.0897* | *0.6072* | *0.0895* | *0.6019* | *0.0994* |

## 3.3 Ablation Study (RQ2)

In the ablation study section, we mainly analyze the Transformer-based encoder and personalized knowledge adaptation unit. We also test the different ways of injecting the knowledge into a recommendation model.

*3.3.1 Transformer-based Knowledge Encoder.* We conduct extensive experiments to verify the effectiveness of the proposed knowledge encoder. The first compared method is removing the Transformer structure in Figure 4 ("w/o TRM"). The second compared method is replacing the Transformer into a feedforward network ("MLP") that takes each feature vector $\{e_i\}_{i=1}^F$ as input separately and output a vector with the same dimension The output of these two methods correspond to $\{g_1^u, ..., g_{F_u}^u, g_1^v, ..., g_{F_v}^v, g_1^c, ..., g_{F_c}^c\}$ in Figure 4. The results are shown in Table 3.

From the results, we could verify the importance of the proposed Transformer-based encoder. The two compared methods do not model the interactions between the input features thus perform

**Table 3: The performance of using different knowledge encoders.**

| Dataset | Encoder | DeepFM | DIN | DCNv2 |
|---------|---------|--------|-----|-------|
| AD | w/o TRM | 0.6281±0.0001 | 0.6244±0.0003 | 0.6278±0.0003 |
| | MLP | 0.6307±0.0002 | 0.6325±0.0003 | 0.6331±0.0002 |
| | D2K-adp-share | **0.6318±0.0004** | **0.6340±0.0003** | **0.6341±0.0004** |
| Eleme | w/o TRM | 0.6011±0.0071 | 0.6012±0.0121 | 0.6076±0.0081 |
| | MLP | 0.6030±0.0023 | 0.6097±0.0105 | 0.6100±0.0051 |
| | D2K-adp-share | **0.6033±0.0069** | **0.6107±0.0106** | **0.6147±0.0088** |

not as well as the Transformer encoder. As the number of input features differs in the training ($F$ features) and inference (3 features) process of the encoder, Transformer is the most suitable structure.

*3.3.2 Personalized Knowledge Adaptation Unit.* To verify the effectiveness of the proposed knowledge adaptation method in Section 2.4.3, we develop four different variants of D2K implementation as shown in Table 1 and Table 2. By comparing the results of the different variants, we have the following observations: (1) D2K-base does not use the adaptation unit. Thus it performs worse than the other three variants in most cases. This result shows that the adaptation unit is essential to the performance, and we have to make the global knowledge adaptive to the current target sample. (2) D2K-adp-sep utilizes a separate embedding table for the input $x$ in the adaptation unit to avoid interference from the original input. D2K-adp-small uses a smaller embedding size than D2K-adp-sep to reduce the number of additional parameters introduced by the adaptation unit. By comparing the results of D2K-adp-sep/D2K-adp-small and D2K-adp-share, we cannot say for sure that incorporating a separate embedding is beneficial. Even if separate embedding is good for performance, it will cost much more GPU memory consumption than the shared embedding variant. Thus we believe using shared embedding is a better practice for the personalized knowledge adaptation unit.

*3.3.3 Knowledge Injection Variants.* We compare three different knowledge injection variants that have been mentioned in Section 2.4.4. The results are shown in Figure 6. "TOWER:LR" and "TOWER:MLP" represent using linear layer and deep neural networks as the additional predictor $\mathcal{P}$ in Eq. (14), respectively.

From the figure, we have the following observations: (1) Concat and additional tower do not necessarily better than the other one. For different datasets, we need to try both of the variants of injecting knowledge. (2) "TOWER:MLP" always performs worse than "TOWER:LR". This could be because the knowledge vectors we use are already mapped into the linearly separable space, as shown in Figure 4. LR is enough to utilize the information in the knowledge vectors, but MLP may be rather overfitting because of its complexity.

## 3.4 Reducing Knowledge Base Size (RQ3)

The major concern of utilizing D2K is its space complexity issue because there could be a large number of unique ternary cross features in real-world applications. To reduce the number of entries, we could omit some of the features to build the knowledge base, such as user ID or item ID, because these features may have too

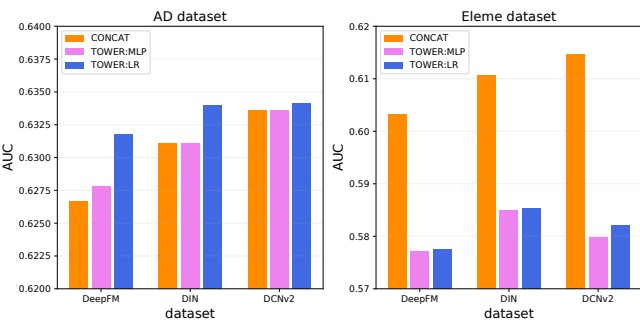

**Figure 6: Comparison between different ways of knowledge injection.**

many values. We further investigate what the performance would be if we decreased the size of the knowledge base of D2K. We select fewer feature fields to reduce the total number of entries in the knowledge base. For example, if the data sample $x$ originally has 20 fields, we only use 18 or 15 fields in the process of building knowledge and querying. The experimental results are shown in Figure 7. We use three different feature sets (FS) with different numbers of selected fields. In the AD dataset, FS1 has 36 ternary cross-feature fields, FS2 has 8 cross fields, and FS3 has 3 cross fields. The numbers for the Eleme dataset are 36, 12 and 8. Details are in Appendix A.5.

We plot the number of entries in each feature set (histogram) with the corresponding AUCs (blue curve) in each sub-figure. We further show the AUCs of the Fixed Window (R) on $\mathcal{D}_{tr}$ (red horizontal line) and the best baseline of each group of the experiment (orange horizontal line).

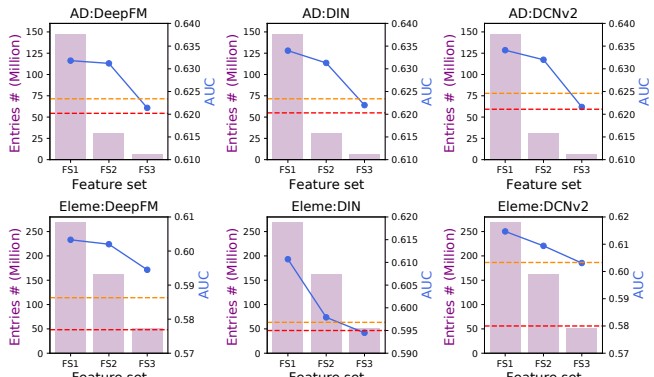

**Figure 7: Performance of D2K under different sizes of knowledge base. The orange line represents AUC of the best baseline of each group and the red line represents the AUC of the Fixed Window (R).**

From the figure, we have the following analysis: (1) The number of entries will be largely reduced if we use fewer fields to build the knowledge base. In the AD dataset, FS2 & FS3 knowledge bases only have 21% and 4% entries of FS1. While in Eleme, the numbers are 60% and 19%. (2) Reducing the number of entries will no doubt hurt the performance of D2K on each backbone model. But we could observe a marginal effect of increasing the size of the knowledge base in most cases. For example, in AD:DeepFM, the FS1 knowledge

base is about five times FS2, while FS2 is about five times FS3. But FS2's AUC has more improvement against FS3 than FS1 against FS2. (3) Although the performance of D2K drops significantly with the smaller knowledge base, it could still outperform the basic baseline of Fixed Window (R) on $\mathcal{D}_{tr}$, which is widely used in real-world systems. Furthermore, when compared with the best baseline of each group, D2K with the smallest knowledge base still achieves competitive performance. These results demonstrate the practicability of D2K and verify the effectiveness of reducing the size of the knowledge base by using fewer features fields.

## 3.5 Knowledge Update (RQ4)

In this section, we investigate the performance of using different update mechanism described in Section 2.3.4. We split $\mathcal{D}_{old}$ into two parts chronologically: the first is used in knowledge base initialization and the second is used to update the base, as described in Section 2.3.4.

The results are shown in Table 4. We have the following observations: (1) The performance of different update mechanisms is close to "w/o update". This is because the two $\mathcal{D}_{old}$ we use covers 4 days, thus one single knowledge encoder may be sufficient enough to memorize the useful patterns. (2) The average pooling (AP) strategy is better than the recent priority (RP) method. Because the recent priority method simply drops the old knowledge vector, which may result in information loss although it is easier to implement. The average pooling strategy is like an "ensemble" of the old and new encoder models, thus it has better performance.

We are also curious about the situation if the knowledge is not updated timely. Thus we test the robustness against the outdated knowledge in Appendix A.6.

**Table 4: The performance of using different update mechanisms. RP: Recent Priority, AP: Average Pooling.**

| Dataset | Update | DeepFM | DIN | DCNv2 |
|---------|--------|--------|-----|-------|
| AD | w/o update | 0.6318±0.0004 | 0.6340±0.0003 | 0.6341±0.0004 |
| | RP | 0.6309±0.0002 | 0.6322±0.0002 | 0.6333±0.0003 |
| | AP | **0.6322±0.0002** | **0.6343±0.0003** | **0.6344±0.0001** |
| Eleme | w/o update | 0.6033±0.0069 | 0.6107±0.0106 | 0.6147±0.0088 |
| | RP | 0.6030±0.0073 | 0.6009±0.0118 | 0.6110±0.0074 |
| | AP | **0.6039±0.0055** | **0.6118±0.0099** | **0.6151±0.0076** |

## 4 RELATED WORKS

This section reviews related works on continual learning with and without external memory and coreset selection techniques.

**Continual Learning w/o External Memory.** Continual learning aims at continually accumulating knowledge over time without the need to retrain from scratch [7]. Knowledge is gradually integrated into the model parameters by managing the stability-plasticity dilemma. Wang et al. [28] propose incrementally updating the model parameters with only the newly incoming samples since the last model update. The old model acts as the teacher to distill knowledge to the new model. Mi et al. [14] propose to periodically replay previous training samples to the current model with an adaptive distillation loss. Conure [31] supports the continual learning for

multiple recommendation tasks by isolating the model parameters for different tasks. Zhang et al. [32] propose to use meta-learning in continual model updates for recommendation by learning to optimize future performance.

Preserving the knowledge in the parameters of the models suffers from catastrophic forgetting. D2K, on the contrary, has better flexibility and scalability by maintaining the knowledge base.

**Continual Learning w/ External Memory.** Memory-augmented networks [9, 12] could be viewed as a case of continual learning with external memory to preserve additional knowledge. In the recommendation scenarios, the user behavior data is used to update the states of the external memory, storing the user representation vectors. NMRN [26] maintains an augmented memory of several latent vectors activated by a specific user ID. NMRN reads from memory and generates a vector representing the user's long- short-term interests. RUM [4] introduces a first-in-first-out writing mechanism to emphasize the latest user behaviors in the user memory matrix. KSR [11] constructs a key-value memory network to store the user's attribute-level preferences by incorporating a knowledge graph. To handle the lifelong user modeling, Ren et al. [22] further adopts a hierarchical and periodical updating mechanism to capture the user's multi-scale interests in their memory. Similarly, user representation vectors are also maintained in the external memory for UIC [17], in which the computation of the user representation vectors is decoupled from the inference process to reduce the latency.

However, the keys used in the memory-augmented networks are unary (e.g., indexed by user ID, in most cases). The extracted knowledge is insufficient to preserve the abundant knowledge in a user log. Moreover, the maintenance of the memory-augmented networks is always complicated [18] because the stored vectors will be updated in the training process as parameters thus should always be maintained in GPUs. D2K's knowledge vectors, however, is constant values that could be stored in CPU memories and persisted in hard disks.

**Coreset Selection.** The coreset is a subset of data samples on which the model could achieve comparable performance as trained on the full dataset, which is commonly seen in the field of Computer Vision [3, 5]. SVP-CF [23] designs a data-specific sampling strategy for recommendation, which employs a base model as a proxy to tag the importance of each data sample. The importance of each sample is defined as the average prediction error of the proxy model over epochs. Coreset selection is generally heuristic and lacks generalizability, where the sampling strategy relies heavily on the selected proxy or pre-defined rules.

## 5 CONCLUSIONS AND FUTURE WORKS

This paper proposes the D2K framework to turn historical data into retrievable knowledge effectively. We designed a ternary knowledge base that uses user-item-context cross features as keys. The ternary knowledge preserves the abundant information extracted from the user logs. D2K shows superiority over the other methods on two large-scale datasets. For future work, we plan to support fussy search. Now D2K uses an exact match on the ternary features, which may cause some of the queries not to have a match in the knowledge base and it may hurt the performance.

# A APPENDIX

## A.1 Pseudo Code of Knowledge Base Generation Process in Section 2.3

The algorithm of knowledge base generation is shown in Algorithm 1.

---

**Algorithm 1** Knowledge base generation of D2K.

---

**Require:** Old dataset $\mathcal{D}_{old}$, knowledge encoder Enc.
**Ensure:** The knowledge base $\mathcal{K}$.

1: Train the encoder Enc on $\mathcal{D}_{old}$ using loss function in Eq. (7).
2: **repeat**
3:    Get all of the ternary user-item-context cross features of sample $x$ as $\{\langle f_i^u, f_j^v, f_k^c \rangle\}$.
4:    **repeat**
5:       Feed the cross feature $\langle f_i^u, f_j^v, f_k^c \rangle$ into Enc and get the resulted knowledge vector $z_{ijk}$.
6:       Add the entry to the knowledge base as $\mathcal{K}[\langle f_i^u, f_j^v, f_k^c \rangle] = z_{ijk}$
7:    **until** all the ternary features of sample $x$ are processed.
8: **until** all samples in $\mathcal{D}_{old}$ have been processed.

---

## A.2 Datasets Statistics

Unlike most of the recommendation datasets that only record the clicked (positive) samples, these two datasets also include the exposed but not clicked (negative) samples so that no negative sampling process is required. The statistics of the datasets are shown in Table 5.

**Table 5: The preprocessed dataset statistics. "ID #" represents the number of unique feature ids.**

| Dataset | Users # | Items # | Interaction # | Feature Fields # | ID # |
|---|---|---|---|---|---|
| AD | 1,061,768 | 827,009 | 25,029,435 | 14 | 3,029,333 |
| Eleme | 5,782,482 | 1,853,764 | 49,114,930 | 25 | 16,516,885 |

## A.3 Time and Space Overhead of D2K

We list the time overhead of the retrieval process, the knowledge building process, and the memory consumption of loading the D2K knowledge base, as shown in Table 6. We tested these statistics on a machine with an AMD EPYC 7302 16-Core Processor as CPU and 256GB memory. The additional overhead introduced by D2K is generally acceptable for real-world deployment. Specifically, the retrieval time for each batch (1024 samples) is less than 100ms, which meets the low latency requirement of the recommender systems [17, 25]. The knowledge base only needs to be built once for later use with reasonable memory consumption and wall time.

## A.4 Direct Knowledge

As we define the knowledge of D2K as "direct" in Definition 1, we test the performance of solely utilizing the knowledge without the input $x$ being fed to the recommendation model. We only utilize the

**Table 6: Time & space overheads of D2K.**

| Dataset | Time of Retrieval | Time of Building $\mathcal{K}$ | Mem. Consumption |
|---|---|---|---|
| AD | 44.7ms/batch | 62mins | 76.32GB |
| Eleme | 89.8ms/batch | 113mins | 138.48GB |

linear knowledge tower as the predictor to produce the estimation, which is the $\mathcal{P}(\text{concat}(\{\hat{z}_{ijk}\}))$ part in Eq. (14).

"ONLY_KLG_LR w/ adp" uses the adapted knowledge $\hat{z}_{ijk}$ in the linear predictor and "ONLY_KLG_LR w/o adp" uses the global knowledge vector $z_{ijk}$. We also list the performance of the DeepFM trained on $\mathcal{D}_{tr}$ as a basic comparison. The results are shown in Table 7. We observe that direct knowledge with adaptation performs well. It even achieves better results on AD compared to DeepFM. The adaptation unit uses the information of target sample $x$ after all. Thus the performance of "w/o adp" is worse than that of "w/ adp". However, the AUC of "w/o adp" is still better than 0.5 by far, which verifies that the global direct knowledge retrieved by $x$ has considerable discrimination ability.

**Table 7: The performance of using direct knowledge solely. "adp": personalized knowledge adaptation unit.**

| Dataset | Model | AUC | LL |
|---|---|---|---|
| AD | DeepFM trained on $\mathcal{D}_{tr}$ | 0.6202±0.0009 | 0.1952±0.0002 |
| | ONLY_KLG_LR w/ adp | 0.6247±0.0003 | 0.1943±0.0002 |
| | ONLY_KLG_LR w/o adp | 0.6068±0.0002 | 0.1955±0.0000 |
| Eleme | DeepFM trained on $\mathcal{D}_{tr}$ | 0.5769±0.0026 | 0.0987±0.0107 |
| | ONLY_KLG_LR w/ adp | 0.5663±0.0012 | 0.0972±0.0031 |
| | ONLY_KLG_LR w/o adp | 0.5426±0.0007 | 0.0901±0.0000 |

## A.5 Feature Selection Details in Section 3.4

The feature selection details are shown in Table 8. The corresponding feature meanings could be found in the dataset links in Section 3.1.1.

## A.6 Outdated Knowledge

As the methods in Section 3.1.2 need to preserve the knowledge in $\mathcal{D}_{old}$, we are curious about the robustness of the stored knowledge. We test how the methods will perform if the knowledge preserved by them is outdated. In the real-world, it is possible that the knowledge is not updated timely online.

To implement the outdated knowledge, we intentionally left a gap between $\mathcal{D}_{old}$ and $\mathcal{D}_{tr}$. It means $\mathcal{D}_{old} = \{\mathcal{D}_1, ..., \mathcal{D}_{p_1}\}$ but $\mathcal{D}_{tr} = \{\mathcal{D}_{p_1+G}, ..., \mathcal{D}_{p_2}\}$, where $G$ is the gap time interval (notations refer to Section 3.1.1). $G$ is set to 24 hours.

The results are shown in Table 9 and Table 10. Every method is tested with outdated knowledge, except for the Fixed Window (R) on $\mathcal{D}_{tr}$ which is used as a basic baseline (The results are the same with Table 1 and Table 2). From the tables, we observe that the performance of all the methods drops because the knowledge is not up-to-date. However, D2K-adp-share still outperforms the basic baseline Fixed Window (R) and other baselines with outdated knowledge. These results show that even if the knowledge is outdated in D2K's knowledge base, a competitive performance could

### Table 8: Detailed feature selection for Section 3.4

| Feature Set | User | Item | Context |
|---|---|---|---|
| (AD) FS1 | userid, cms_segid, cms_group_id, final_gender_code, age_level, shopping_level, occupation, hist_brands, hist_cates | adgroup_id, cate_id, campaign_id, customer | weekday |
| (AD) FS2 | userid, cms_segid, cms_group_id, hist_cates | adgroup_id, cate_id | weekday |
| (AD) FS3 | cms_segid, cms_group_id, hist_brands | cate_id | weekday |
| (Eleme) FS1 | user_id, gender, visit_city, is_supervip, shop_id_list, category_1_id_list | shop_id, city_id, district_id, brand_id, category_1_id, merge_standard_food_id | hours |
| (Eleme) FS2 | user_id, gender, shop_id_list | shop_id, city_id, district_id, brand_id | hours |
| (Eleme) FS3 | gender, visit_city, is_supervip | brand_id, category_1_id | hours |

still be expected. The robustness of D2K is further verified compared to other methods to preserve knowledge.

### Table 9: The performance of outdated knowledge on AD.

| Method | DeepFM | DIN | DCNv2 |
|---|---|---|---|
| Fixed Window (R) | 0.6202±0.0009 | 0.6203±0.0008 | 0.6211±0.0011 |
| Incremental | 0.6182±0.0012 | 0.6181±0.0016 | 0.6190±0.0012 |
| IncCTR(KD-batch) | 0.6188±0.0003 | 0.6185±0.0005 | 0.6114±0.0009 |
| Pretrain embedding | 0.6111±0.0008 | 0.6144±0.0005 | 0.6158±0.0009 |
| MemoryNet | 0.6110±0.0009 | 0.6138±0.0008 | 0.6159±0.0006 |
| Random | 0.6194±0.0006 | 0.6206±0.0003 | 0.6210±0.0004 |
| SVP-CF | 0.6191±0.0010 | 0.6206±0.0003 | 0.6213±0.0003 |
| D2K-adp-share | **0.6228±0.0011** | **0.6262±0.0002** | **0.6253±0.0021** |

### Table 10: The performance of outdated knowledge on Eleme.

| Method | DeepFM | DIN | DCNv2 |
|---|---|---|---|
| Fixed Window (R) | 0.5769±0.0026 | 0.5950±0.0125 | 0.5800±0.0054 |
| Incremental | 0.5797±0.0091 | 0.5751±0.0103 | 0.5829±0.0081 |
| IncCTR(KD-batch) | 0.5867±0.0045 | 0.6065±0.0059 | 0.6098±0.0124 |
| Pretrain embedding | 0.5661±0.0029 | 0.5813±0.0025 | 0.5733±0.0110 |
| MemoryNet | 0.5681±0.0092 | 0.5909±0.0116 | 0.5865±0.0112 |
| Random | 0.5741±0.0128 | 0.5957±0.0094 | 0.5905±0.0089 |
| SVP-CF | 0.5813±0.0129 | 0.5833±0.0066 | 0.5854±0.0082 |
| D2K-adp-share | **0.6028±0.0102** | **0.6079±0.0074** | **0.6123±0.0081** |

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
