# OpenReview forum: "D2K: Turning Historical Data into Retrievable Knowledge for Recommender Systems"
_ACM.org/TheWebConf/2025/Conference — WWW 2025 Oral_

### Official Review · Reviewer_8dRV · 2024-11-22

**Novelty:** 4
**Technical Quality:** 6

**Review:**

This paper focuses on the problem of integrating old knowledge with new data in recommendation. It proposes a data-centric framework, D2K, to model old data as a knowledge base.

This is more flexible and scalable than parameter-centric frameworks. When new data comes, old knowledge is available to be queried. This paper compares D2K with baselines that cover three categories of the related works (Continual Learning, External Memory, Coreset Selection). Analysis in the experimental sections demonstrates some insights. For example, outdated knowledge is also analyzed. So I give 6 points for technical quality.

However, the architecture proposed seems complex.  Please see the questions below.

**Questions:**

1. Could you explain more about the motivation for each component of model design, especially for the knowledge adaptation unit?
2. "The parameter-centric methods transform the raw data into a fixed number of parameters and thereby cannot scale up as the data size
increases, resulting in the loss of information. " in the introduction section: I don't quite agree with this viewpoint.

**Reviewer Confidence:**

2: The reviewer is willing to defend the evaluation, but it is likely that the reviewer did not understand parts of the paper

**Scope:**

4: The work is relevant to the Web and to the track, and is of broad interest to the community

---

### Official Review · Reviewer_tCVE · 2024-11-30

**Novelty:** 4
**Technical Quality:** 4

**Review:**

Pros

1.	The paper presents a novel data-centric approach that leverages a ternary knowledge base, capturing user-item-context interactions, and employs a modular design that facilitates seamless integration with a wide range of recommendation models, thereby offering both flexibility and adaptability for diverse applications.

2.	Comprehensive experiments are conducted to rigorously validate the proposed framework across multiple datasets, featuring robust comparisons against various baseline methods and detailed ablation studies, which collectively underscore the critical role of each component in achieving superior performance and ensuring the framework's robustness.

3.	The approach demonstrates significant improvements in key metrics such as AUC, reflecting enhanced ranking quality, and log loss, indicating better predictive accuracy. These results highlight its strong potential for real-world deployment, particularly in applications requiring high scalability and impactful online performance improvements.

Cons

1.	The limited dataset diversity, with experiments primarily focused on e-commerce platforms such as AD and Eleme, raises concerns about the framework's generalizability and effectiveness in other domains, such as social media, video streaming, or personalized content recommendations, where user behavior and contextual dynamics may differ significantly.

2.	While the paper emphasizes scalability as one of its strengths, it lacks quantitative benchmarks to validate claims related to storage efficiency, retrieval latency, and computational overhead on large-scale datasets, which are critical factors for evaluating the framework's practicality in real-world, high-volume applications.

3.	The model framework diagram, though providing a general overview, appears overly simplistic and could be enhanced with additional layers of detail, especially in the Knowledge Injection section. The absence of a clear depiction of specific processes, such as how knowledge vectors are integrated with the recommendation model or how adaptation mechanisms operate, may obscure the comprehensive understanding of this critical component.

**Questions:**

1.Could you elaborate on how the "Recent Priority (RP)" and "Average Pooling (AP)" strategies scale with increasing data sizes?

2.Why you choose the Transformer architecture over alternatives like graph-based encoders or convolutional networks?  Have you explored the impact of deeper or more complex Transformers on encoding performance?

3.Why is it possible to achieve Personalized Knowledge Adaptation solely through the use of different MLP layers?  Could you provide more detailed and specific explanations about the implementation methods for the four different variants of the D2K framework?

**Reviewer Confidence:**

3: The reviewer is confident but not certain that the evaluation is correct

**Scope:**

4: The work is relevant to the Web and to the track, and is of broad interest to the community

---

### Official Review · Reviewer_bw68 · 2024-12-02

**Novelty:** 5
**Technical Quality:** 5

**Review:**

This paper proposes a data-centric framework called D2K (Data-to-Knowledge), which transforms massive user behavior data into retrievable knowledge. Unlike previous methods that store unary knowledge (such as user or item information), D2K stores ternary knowledge, considering user, item, and context for more effective recommendations. The framework uses a Transformer-based knowledge encoder to convert old data into cross-feature knowledge, and a personalized knowledge adaptation unit to tailor this knowledge for specific recommendations.

S1. This paper addresses the scalability and flexibility challenges posed by massive user historical data. The problem is significant, and the motivation for tackling it is strong.

S2. The paper is well-written and easy to follow.

S3. The experiment design is detailed.

O1. The paper only evaluates three backbone models in the experiments. It would be beneficial to include a broader range of backbones for a more comprehensive analysis.

O2. Only two datasets are included in the experiment section.

**Questions:**

1.	The rationale for selecting the three backbones seems to be based on their popularity as widely used models. However, would it be possible to consider recommendation models as backbones from perspectives beyond just popularity.

2.	There are several well-known methods for addressing the catastrophic forgetting problem, such as “Continual Test-Time Domain Adaptation” and “NOTE: Robust Continual Test-Time Adaptation Against Temporal Correlation.” How do the authors position D2K in addressing the catastrophic forgetting problem compared to these methods?

**Reviewer Confidence:**

3: The reviewer is confident but not certain that the evaluation is correct

**Scope:**

4: The work is relevant to the Web and to the track, and is of broad interest to the community

---

### Official Review · Reviewer_xtp4 · 2024-12-02

**Novelty:** 6
**Technical Quality:** 6

**Review:**

The paper introduces a data-centric approach to store knowledge (transformer-based encoder) from historical data and proposes a personalized knowledge adaptation unit to integrate retrieved knowledge with a recent recommendation model trained on recent data. This combination aims to enhance model performance effectively. Experiments conducted on various datasets validate the approach's effectiveness.

- **Pros**
	- The idea is innovative and well-grounded.
	- The paper is easy to comprehend and follow.
	- Exhaustive experiments provide strong evidence for its effectiveness.

- **Cons**
	- The number of keys (ternary information) grows significantly with the increase in user features, item features, and contextual data. This raises concerns about scalability, particularly in terms of storage (space) and retrieval (search) costs.

**Questions:**

- Would it be possible to analyze the complexity of the proposed approach in the main content? This would address concerns regarding computational efficiency.
- Why does the fixed window (A) approach perform worse than the proposed method, even though it is trained on both historical and recent data?

**Reviewer Confidence:**

3: The reviewer is confident but not certain that the evaluation is correct

**Scope:**

4: The work is relevant to the Web and to the track, and is of broad interest to the community